# INTEGER NETWORKS FOR DATA COMPRESSION WITH LATENT-VARIABLE MODELS

**Johannes Ballé, Nick Johnston & David Minnen**
Google
Mountain View, CA 94043, USA
{jballe,nickj,dminnen}@google.com

## ABSTRACT

We consider the problem of using variational latent-variable models for data compression. For such models to produce a compressed binary sequence, which is the universal data representation in a digital world, the latent representation needs to be subjected to entropy coding. Range coding as an entropy coding technique is optimal, but it can fail catastrophically if the computation of the prior differs even slightly between the sending and the receiving side. Unfortunately, this is a common scenario when floating point math is used and the sender and receiver operate on different hardware or software platforms, as numerical round-off is often platform dependent. We propose using integer networks as a universal solution to this problem, and demonstrate that they enable reliable cross-platform encoding and decoding of images using variational models.

## 1 INTRODUCTION

The task of information transmission in today's world is largely divided into two separate endeavors: *source coding*, or the representation of data (such as audio or images) as sequences of bits, and *channel coding*, representing sequences of bits as analog signals on imperfect, physical channels such as radio waves (Cover and Thomas, 2006). This decoupling has substantial benefits, as the binary representations of arbitrary data can be seamlessly transmitted over arbitrary physical channels by only changing the underlying channel code, rather than having to design a new code for every possible combination of data source and physical channel. Hence, the universal representation of any compressed data today is the binary channel, a representation which consists of a variable number of binary symbols, each with probability $\frac{1}{2}$, and no noise (i.e. uncertainty).

As a latent representation, the binary channel unfortunately is a severe restriction compared to the richness of latent representations defined by many variational latent-variable models in the literature (e.g., Kingma and Welling, 2014; Sønderby et al., 2016; van den Oord et al., 2017), and in particular models targeted at data compression (Theis et al., 2017; Ágústsson et al., 2017; Ballé et al., 2018). Variational latent-variable models such as VAEs (Kingma and Welling, 2014) consist of an encoder model distribution $e(\boldsymbol{y} \mid \boldsymbol{x})$ bringing the data $\boldsymbol{x}$ into a latent representation $\boldsymbol{y}$, and a decoder model distribution $d(\boldsymbol{x} \mid \boldsymbol{y})$, which represents the data likelihood conditioned on the latents. Given an encoder $e$, we observe the marginal distribution of latents $m(\boldsymbol{y}) = \mathbb{E}_{\boldsymbol{x}}[e(\boldsymbol{y} \mid \boldsymbol{x})]$, where the expectation runs over the (unknown) data distribution. The prior $p(\boldsymbol{y})$ is a variational estimate of the marginal (Alemi et al., 2018).

By choosing the parametric forms of these distributions and the training objective appropriately, many such models succeed in representing relevant information in the data they are trained for quite compactly (i.e., with a small expected Kullback–Leibler (KL) divergence between the encoder and the prior, $\mathbb{E}_{\boldsymbol{x}} D_{\mathrm{KL}}[e\|p]$), and so may be called *compressive* in a sense. However, not all of them can be directly used for practical data compression, as the representation needs to be further converted into binary (entropy encoded). This conversion is typically performed by *range coding*, or arithmetic coding (Rissanen and Langdon, 1981). Range coding is asymptotically optimal: the length of the binary sequence quickly converges to the expected KL divergence in bits, for reasonably large sequences (such as, for one image). For this to hold, the following requirements must be satisfied:

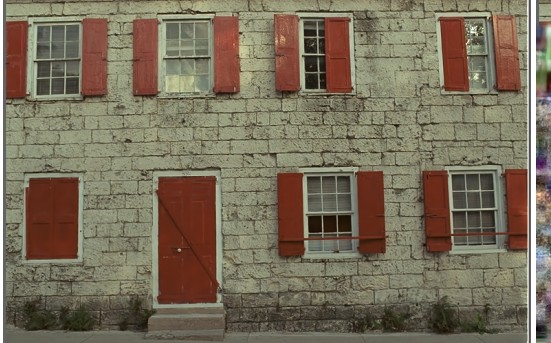 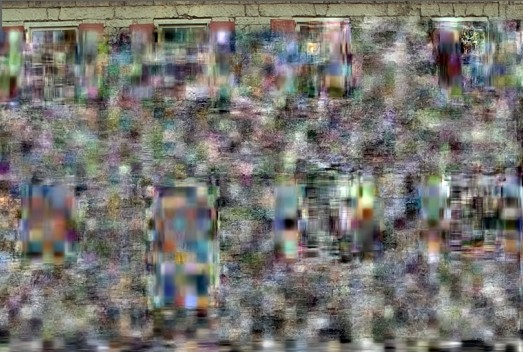

Figure 1: The same image, decoded with a model computing the prior using integer arithmetic (left), and the same model using floating point arithmetic (right). The image was decoded correctly, beginning in the top-left corner, until floating point round-off error caused a small discrepancy between the sender's and the receiver's copy of the prior, at which point the error propagated catastrophically.

- The representation must be discrete-valued, i.e. have a finite number of states, and be noiseless – i.e. the conditional entropy of the encoder must be zero:

$$H[e] = \mathbb{E}_{\boldsymbol{x}} \, \mathbb{E}_{\boldsymbol{y} \sim e}[-\log e(\boldsymbol{y} \mid \boldsymbol{x})] = 0. \tag{1}$$

- All scalar elements of the representation $\boldsymbol{y}$ must be brought into a total ordering, and the prior needs to be written using the chain rule of calculus (as a product of conditionals), as the algorithm can only encode or decode one scalar random variable at a time.

- Both sides of the binary channel (i.e. sender and receiver) must be able to evaluate the prior, and they must have identical instances of it.

The latter point is crucial, as range coding is extremely sensitive to differences in $p$ between sender and receiver – so sensitive, in fact, that even small perturbations due to floating point round-off error can lead to catastrophic error propagation. Unfortunately, numerical round-off is highly platform dependent, and in typical data compression applications, sender and receiver may well employ different hardware or software platforms. Round-off error may even be non-deterministic on one and the same computer. Figure 1 illustrates a decoding failure in a model which computes $p$ using floating point math, caused by such computational non-determinism in sender vs. receiver. Recently, latent-variable models have been explored that employ artificial neural networks (ANNs) to compute hierarchical or autoregressive priors (Sønderby et al., 2016; van den Oord et al., 2017), including some of the best-performing learned image compression models (Ballé et al., 2018; Minnen et al., 2018; Klopp et al., 2018). Because ANNs are typically based on floating point math, these methods are vulnerable to catastrophic failures when deployed on heterogeneous platforms.

To address this problem, and enable use of powerful learned variational models for real-world data compression, we propose to use integer arithmetic in these ANNs, as floating-point arithmetic cannot presently be made deterministic across arbitrary platforms. We formulate a type of quantized neural network we call *integer networks*, which are specifically targeted at generative and compression models, and at preventing computational non-determinism in computation of the prior. Because full determinism is a feature of many existing, widely used image and video compression methods, we also consider using integer networks end to end for computing the representation itself.

## 2 INTEGER NEURAL NETWORKS

ANNs are typically composite functions that alternate between linear and elementwise nonlinear operations. One linear operation followed by a nonlinearity is considered one layer of the network. To ensure that such a network can be implemented deterministically on a wide variety of hardware platforms, we restrict all the data types to be integral, and all operations to be implemented either with basic arithmetic or lookup tables. Because integer multiplications (including matrix multiplications or convolutions) increase the dynamic range of the output compared to their inputs, we introduce an additional step after each linear operator, where we divide each of its output by a learned parameter.

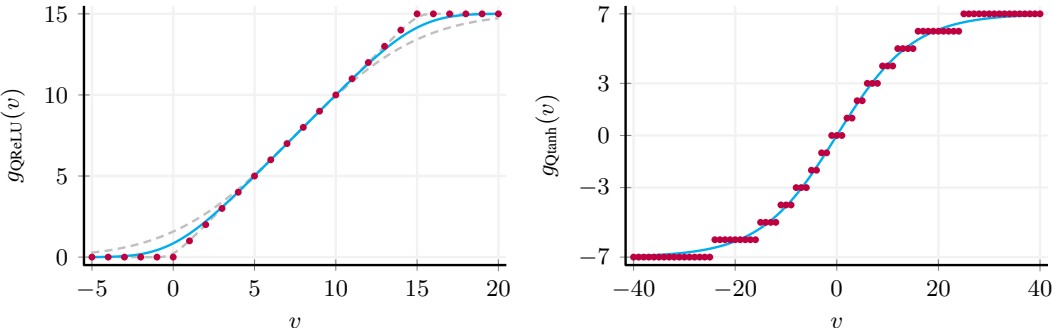

Figure 2: Left: Example nonlinearity implementing a saturating rectifier for 4-bit unsigned integer outputs, given by $g_{\text{QReLU}}(v) = \max(\min(v, 15), 0)$. This nonlinearity can be implemented deterministically either using a lookup table or simply using a clipping operation. The corresponding scaled cumulative of a generalized Gaussian with $\beta = 4$ used for computing gradients is plotted in cyan, and other choices of $\beta$ in gray. Right: Example nonlinearity approximating hyperbolic tangent for 4-bit signed integer outputs, given by $g_{\text{Qtanh}}(v) = Q(7 \tanh(\frac{v}{15}))$. This nonlinearity can be implemented deterministically using a lookup table. The corresponding scaled hyperbolic tangent used for computing gradients is plotted in cyan.

Concretely, we define the relationship between inputs $\boldsymbol{u}$ and outputs $\boldsymbol{w}$ of one layer as:

$$\boldsymbol{v} = (\boldsymbol{H}\boldsymbol{u} + \boldsymbol{b}) \oslash \boldsymbol{c}, \tag{2}$$

$$\boldsymbol{w} = g(\boldsymbol{v}). \tag{3}$$

In order, the inputs $\boldsymbol{u}$ are subjected to a linear transform $\boldsymbol{H}$ (a matrix multiplication, or a convolution); a bias vector $\boldsymbol{b}$ is added; the result is divided elementwise by a vector $\boldsymbol{c}$, yielding an intermediate result vector $\boldsymbol{v}$; and finally, an elementwise nonlinearity $g$ is applied to $\boldsymbol{v}$.

The activations $\boldsymbol{w}$ and all intermediate results, as well as the parameters $\boldsymbol{H}$, $\boldsymbol{b}$, and $\boldsymbol{c}$ are all defined as integers. However, they may use differing number formats. For $\boldsymbol{v}$ to be integral, we define $\oslash$ here to perform rounding division (equivalent to division followed by rounding to the nearest integer). In programming languages such as C, this can be implemented with integer operands $m, n$ as

$$m \oslash n = Q(\tfrac{m}{n}) = (m + n \mathbin{/\!/} 2) \mathbin{/\!/} n, \tag{4}$$

where $Q$ rounds to the nearest integer and $/\!/$ is floor division; here, the addition can be folded into the bias $\boldsymbol{b}$ as an optimization. We constrain the linear filter coefficients $\boldsymbol{H}$ and the bias vector $\boldsymbol{b}$ to generally use signed integers, and the scaling vector $\boldsymbol{c}$ to use unsigned integers. We implement the accumulators of the linear transform with larger bit width than the activations and filter coefficients, in order to reflect the potentially increased dynamic range of multiplicative operations. We assume here that the bias and scaling vectors, as well as the intermediate vector $\boldsymbol{v}$, have the same bit width as the accumulators.

The elementwise nonlinearity $g$ must be saturating on both ends of its domain, because integers can only represent finite number ranges. In order to maximize utility of the dynamic range, we scale nonlinearities such that their range matches the bit width of $\boldsymbol{w}$, while their domain can be scaled somewhat arbitrarily. Depending on the range of the nonlinearity, the activations $\boldsymbol{w}$ may use a signed or unsigned number format. For instance, a reasonable choice of number formats and nonlinearity would be:

$$\boldsymbol{H} : \text{8-bit signed}$$
$$\boldsymbol{b}, \boldsymbol{v} : \text{32-bit signed (same as accumulator)}$$
$$\boldsymbol{c} : \text{32-bit unsigned}$$
$$\boldsymbol{w} : \text{8-bit unsigned}$$
$$g_{\text{QReLU}}(v) = \max(\min(v, 255), 0)$$

In this example, the nonlinearity can be implemented with a simple clipping operation. Refer to figure 2, left, for a visualization (for visualization purposes, the figure shows a smaller bit width).

Another example is:

$$\boldsymbol{H} : \text{4-bit signed}$$
$$\boldsymbol{b}, \boldsymbol{v} : \text{16-bit signed (same as accumulator)}$$
$$\boldsymbol{c} : \text{16-bit unsigned}$$
$$\boldsymbol{w} : \text{4-bit signed}$$
$$g_{\text{Qtanh}}(v) = Q\big(7\tanh(\tfrac{v}{15})\big)$$

Here, the nonlinearity approximates the hyperbolic tangent, a widely used nonlinearity. It may be best implemented using a lookup table (see figure 2, right, for a visualization). We scale its range to fill the 4-bit signed integer number format of $\boldsymbol{w}$ by multiplying its output with 7. The domain can be scaled somewhat arbitrarily, since $\boldsymbol{v}$ has a larger bit width than $\boldsymbol{w}$. When it is chosen too small, $\boldsymbol{w}$ may not utilize all integer values, leading to a large quantization error. When it is chosen too large, overflow may occur in $\boldsymbol{v}$, or the size of the lookup table may grow too large for practical purposes. Therefore, it is best to determine the input scaling based on the shape of the nonlinearity and the available dynamic range. Here, we simply chose the value of 15 "by eye", so that the nonlinearity is reasonably well represented with the lookup table (i.e., we made sure that at least two or three input values are mapped to each output value, in order to preserve the approximate shape of the nonlinearity).

## 3 TRAINING INTEGER NEURAL NETWORKS

To effectively accumulate small gradient signals, we train the networks entirely using floating point computations, rounded to integers after every computational operation, while the backpropagation is done with full floating point precision. More concretely, we define the integer parameters $\boldsymbol{H}$, $\boldsymbol{b}$, and $\boldsymbol{c}$ as functions of their floating point equivalents $\boldsymbol{H}'$, $\boldsymbol{b}'$, and $\boldsymbol{c}'$, respectively:

$$\boldsymbol{H} = \begin{bmatrix} Q(\boldsymbol{h}'_1/s(\boldsymbol{h}'_1)) \\ \vdots \\ Q(\boldsymbol{h}'_N/s(\boldsymbol{h}'_N)) \end{bmatrix}, \tag{5}$$

$$\boldsymbol{b} = Q\big(2^K \boldsymbol{b}'\big), \tag{6}$$

$$\boldsymbol{c} = Q\big(2^K r(\boldsymbol{c}')\big). \tag{7}$$

Here, we simply rescale each element of $\boldsymbol{b}'$ using a constant $K$, which is the bit-width of the kernel $\boldsymbol{H}$ (e.g. 8-bits in the QReLu networks), and round it to the nearest integer. The reparameterization mapping $r$ is borrowed from Ballé (2018):

$$r(c') = \max\Big(c', \sqrt{1+\epsilon^2}\Big)^2 - \epsilon^2. \tag{8}$$

When $\boldsymbol{c}$ is small, perturbations in $\boldsymbol{c}$ can lead to excessively large fluctuations of the quotient (i.e., the input to the nonlinearity). This leads to instabilities in training. $r$ ensures that values of $\boldsymbol{c}$ are always positive, while gracefully scaling down gradient magnitudes on $\boldsymbol{c}$ near zero. Effectively, the step size on $\boldsymbol{c}$ is multiplied with a factor that is approximately linear in $\boldsymbol{c}$ (Ballé, 2018).

Before rounding the linear filter coefficients in $\boldsymbol{H}' = [\boldsymbol{h}'_1, \ldots, \boldsymbol{h}'_N]^\top$, we apply a special rescaling function $s$ to each of its filters $\boldsymbol{h}'$:

$$s(\boldsymbol{h}') = \max\Big((-2^{K-1})^{-1}\min_i h'_i, (2^{K-1}-1)^{-1}\max_i h'_i, \epsilon\Big). \tag{9}$$

$s$ rescales each filter such that at least one of its minimum and maximum coefficients hits one of the dynamic range bounds ($-2^{K-1}$ and $2^{K-1}-1$), while keeping zero at zero. This represents the finest possible quantization of the filter given its integer representation, and thus maximizes accuracy. To prevent division by zero, we ensure the divisor is larger than or equal to a small constant $\epsilon$ (for example, $\epsilon = 10^{-20}$).

In order to backpropagate gradient signals into the parameters, one cannot simply take gradients of the loss function with respect to $\boldsymbol{H}'$, $\boldsymbol{b}'$, or $\boldsymbol{c}'$, since the rounding function $Q$ has zero gradients almost everywhere, except for the half-integer positions where the gradient is positive infinity. A

simple remedy is to replace the derivative of $Q$ with the identity function, since this is the smoothed gradient across all rounded values.

Further, we treat the rescaling divisor $s$ as if it were a constant. That is, we compute the derivatives of the loss function with respect to $\boldsymbol{H}'$, $\boldsymbol{b}'$, and $\boldsymbol{c}'$ as with the chain rule of calculus, but overriding:

$$\frac{\partial \boldsymbol{h}}{\partial \boldsymbol{h}'} := \frac{1}{s(\boldsymbol{h}')}, \qquad \frac{\partial \boldsymbol{b}}{\partial \boldsymbol{b}'} := 2^K, \qquad \frac{\partial \boldsymbol{c}}{\partial \boldsymbol{c}'} := 2^K r'(\boldsymbol{c}'), \tag{10}$$

where $r'$ is the replacement gradient function for $r$ as proposed by Ballé (2018). After training is completed, we compute the integer parameters $\boldsymbol{H}$, $\boldsymbol{b}$ and $\boldsymbol{c}$ one more time, and from then on use them for evaluation. Note that further reparameterization of the kernels $\boldsymbol{H}'$, such as Sadam (Ballé, 2018), or of the biases $\boldsymbol{b}'$ or scaling parameters $\boldsymbol{c}'$, is possible by simply chaining reparameterizations.

In addition to rounding the parameters, it is necessary to round the activations. To obtain gradients for the rounding division $\oslash$, we simply substitute the gradient of floating point division. To estimate gradients for the rounded activation functions, we replace their gradient with the corresponding non-rounded activation function, plotted in cyan in figure 2. In the case of QReLU, the gradient of the clipping operation is a box function, which can lead to training getting stuck, since if activations consistently hit one of the bounds, no gradients are propagated back (this is sometimes called the "dead unit" problem). As a remedy, we replace the gradient instead with

$$\frac{\partial g_{\text{QReLU}}(v)}{\partial v} := \exp\Big(-\alpha^\beta \Big| \frac{2v}{2^L - 1} - 1 \Big|^\beta\Big), \tag{11}$$

where $\alpha = \frac{1}{\beta} \Gamma\big(\frac{1}{\beta}\big)$, and $L$ is the bit width of $\boldsymbol{w}$. This function corresponds to a scaled generalized Gaussian probability density with shape parameter $\beta$. In this context, we can think of $\beta$ as a temperature parameter that makes the function converge to the gradient of the clipping operation as $\beta$ goes to infinity. Although this setting permits an annealing schedule, we simply chose $\beta = 4$ and obtained good results. The integral of this function is plotted in figure 2 (left) in cyan, along with other choices of $\beta$ in gray.

## 4 Computing the prior with integer networks

Suppose our prior on the latent representation is $p(\boldsymbol{y} \mid \boldsymbol{z})$, where $\boldsymbol{z}$ summarizes other latent variables of the representation (it may be empty). To apply range coding, we need to impose a total ordering on the elements of $\boldsymbol{y}$ and write it as a chain of conditionals:

$$p(\boldsymbol{y} \mid \boldsymbol{z}) = \prod_i p(y_i \mid \boldsymbol{y}_{:i}, \boldsymbol{z}), \tag{12}$$

where $\boldsymbol{y}_{:i}$ denotes the vector of all elements of $\boldsymbol{y}$ preceding the $i$th. A common assumption is that $p$ is a known distribution, with parameters $\boldsymbol{\theta}_i$ computed by an ANN $g$:

$$p(\boldsymbol{y} \mid \boldsymbol{z}) = \prod_i p(y_i \mid \boldsymbol{\theta}_i) \text{ with } \boldsymbol{\theta}_i = g(\boldsymbol{y}_{:i}, \boldsymbol{z}) \tag{13}$$

We simply propose here to compute $g$ deterministically using an integer network, discretizing the parameters $\boldsymbol{\theta}$ to a reasonable accuracy. If $p(y_i \mid \boldsymbol{\theta}_i)$ itself cannot be computed deterministically, we can precompute all possible values and express it as a lookup table over $y_i$ and $\boldsymbol{\theta}_i$.

As an example, consider the prior used in the image compression model proposed by Ballé et al. (2018), which is a modified Gaussian with scale parameters conditioned on another latent variable:

$$p(\boldsymbol{y} \mid \boldsymbol{z}) = \prod_i \big(\mathcal{N}(0, \sigma_i^2) * \mathcal{U}(-\tfrac{1}{2}, \tfrac{1}{2})\big)(y_i) \text{ with } \boldsymbol{\sigma} = g(\boldsymbol{z}). \tag{14}$$

We reformulate the scale parameters $\boldsymbol{\sigma}$ as:

$$\sigma_i = \exp\Big(\log(\sigma_{\min}) + \frac{\log(\sigma_{\max}) - \log(\sigma_{\min})}{L - 1} \theta_i\Big), \tag{15}$$

where $\boldsymbol{\theta} = g(\boldsymbol{z})$ is computed using an integer network. The last activation function in $g$ is chosen to have integer outputs of $L$ levels in the range $[0, L - 1]$. Constants $\sigma_{\min}$, $\sigma_{\max}$, and $L$ determine the discretized selection of scale parameters used in the model. The discretization is chosen to be logarithmic, as this choice minimizes $\mathbb{E}_{\boldsymbol{x}} D_{\text{KL}}[e \| p]$ for a given number of levels.

During training, we can simply backpropagate through this reformulation, and through $g$ as described in the previous section. After training, we precompute all possible values of $p$ as a function of $y_i$ and $\theta_i$ and form a lookup table, while $g$ is implemented with integer arithmetic.

## 5 COMPUTING THE REPRESENTATION WITH INTEGER NETWORKS

For certain applications, it can be useful not only to be able to deploy a compression model across heterogenous platforms, but to go even further in also ensuring identical reconstructions of the data across platforms. To this end, it can be attractive to make the entire model robust to non-determinism. To use integer networks in the encoder or decoder, one can use the equivalent construction as in (13): define $e$ or $d$ as a known distribution, with parameters computed by an integer network. To allow the use of range coding, we're especially interested in discrete-valued representations here, such as studied in van den Oord et al. (2017), Jang et al. (2017), Theis et al. (2017), Ágústsson et al. (2017), and Ballé et al. (2018), among others. These approaches typically employ biased gradient estimators.

Jang et al. (2017) and Ágústsson et al. (2017) are concerned with producing gradients for categorical distributions and vector quantization (VQ), respectively. In both methods, the representation is found by evaluating an ANN followed by an $\arg\max$ function, while useful gradients are obtained by substituting the $\arg\max$ with a $\mathrm{softmax}$ function. Since $\arg\max$ can be evaluated deterministically in a platform-independent way, and evaluating a $\mathrm{softmax}$ function with rounded inputs is feasible, integer networks can be combined with these models without additional modifications.

Theis et al. (2017) and Ballé et al. (2018) differ mostly in the details of interaction between the encoder and the prior. These two approaches are particularly interesting for image compression, as they scale well: Image compression models are often trained with a rate–distortion objective with a Lagrange parameter $\lambda$, equivalent to $\beta$ in the $\beta$-VAE objective (Higgins et al., 2017; Alemi et al., 2018). Depending on the parameter, the latent representation carries vastly different amounts of information, and the optimal number of latent states in turn varies with that. While the number of latent states is a hyperparameter that needs to be chosen ahead of time in the categorical/VQ case, the latter two approaches can extend it as needed during training, because the latent states are organized along the real line. Further, for categorical distributions as well as VQ, the required dimensionality of the function computing the parameters grows linearly with the number of latent states due to their use of the $\arg\max$ function. In the latter two models, the number of states can grow arbitrarily without increasing the dimensionality of $g$.

Both Theis et al. (2017) and Ballé et al. (2018) use deterministic encoder distributions (i.e. degenerating to delta distributions) during evaluation, but replace them with probabilistic versions for purposes of estimating $\mathbb{E}_{\boldsymbol{x}} D_{\mathrm{KL}}[e\|p]$ during training. Theis et al. (2017) propose to use the following encoder distribution:

$$e(\boldsymbol{y} \mid \boldsymbol{x}) = \mathcal{U}(\boldsymbol{y} \mid Q(g(\boldsymbol{x})) - \tfrac{1}{2}, Q(g(\boldsymbol{x})) + \tfrac{1}{2}), \tag{16}$$

where $\mathcal{U}$ is the uniform distribution and $g$ is an ANN. They replace the gradient of the quantizer with the identity. During evaluation, $\boldsymbol{y} = Q(g(\boldsymbol{x}))$ is used as the representation. Ballé et al. (2018) use the following distribution during training:

$$e(\boldsymbol{y} \mid \boldsymbol{x}) = \mathcal{U}(\boldsymbol{y} \mid g(\boldsymbol{x}) - \tfrac{1}{2}, g(\boldsymbol{x}) + \tfrac{1}{2}), \tag{17}$$

which makes $\boldsymbol{y}$ shift-invariant. During evaluation, they determine the representation as $\boldsymbol{y} = Q(g(\boldsymbol{x}) - \boldsymbol{o})$, where $\boldsymbol{o}$ is a sub-integer offset chosen such that the mode (or, if it cannot be estimated easily, the median) of the distribution is centered on one of the quantization bins.

If $g$ is implemented with integer networks, the latter approach becomes equivalent to the former, because $g$ then inherently computes integer outputs, and this is effectively equivalent to the quantization in (16). However, we've found that training with this construction leads to instabilities, such that the prior distribution never converges to a stable set of parameters. The reason may be that with quantization in $e$, the marginal $m(\boldsymbol{y}) = \mathbb{E}_{\boldsymbol{x}} e(\boldsymbol{y} \mid \boldsymbol{x})$ resembles a piecewise constant function, while the prior $p$ must be forced to be smooth, or $\mathbb{E}_{\boldsymbol{x}} D_{\mathrm{KL}}[e\|p]$ would not yield any useful gradients. Because the prior is a variational approximation of the marginal, this means that the prior must be regularized (which we did not attempt here – we used the nonparametric density model described in Ballé et al. (2018)). On the other hand, when using (17) without quantization, the marginal is typically a smooth density, and the prior can approximate it closely without the need for regularization.

As a remedy for the instabilities, we propose the following trick: We simply use (17) during training, but define the last layer of $g$ without a nonlinearity and with floating point division, such that the representation is

$$e(\boldsymbol{y} \mid \boldsymbol{x}) = \mathcal{U}(\boldsymbol{y} \mid (\boldsymbol{H}\boldsymbol{u} + \boldsymbol{b})/\boldsymbol{c} - \tfrac{1}{2}, (\boldsymbol{H}\boldsymbol{u} + \boldsymbol{b})/\boldsymbol{c} + \tfrac{1}{2}), \tag{18}$$

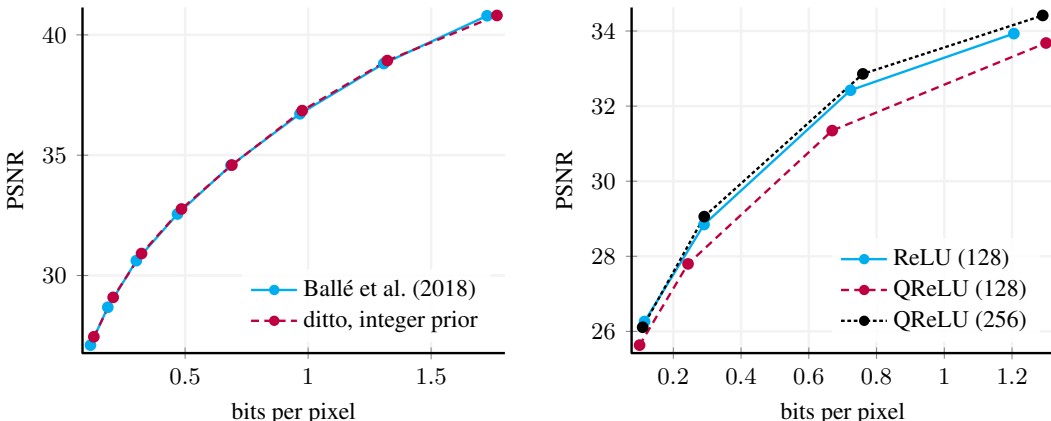

Figure 3: Rate–distortion performance of image compression models with integer priors (left and up is better). Left: performance of Ballé et al. (2018) model vs. the same model with an integer prior. The performance is identical, but the latter can be reliably deployed across different hardware platforms. Right: performance of Ballé (2018) ReLU model with 128 filters per layer vs. the same model, with integer transforms and QReLU activation functions and 128 or 256 filters per layer. The approximation capacity of integer networks is diminished vs. floating point networks, but doubling the number of filters per layer more than compensates for the loss.

| compressed on | CPU 1 | CPU 1 | CPU 1 | CPU 1 | GPU 1 | GPU 1 | GPU 1 | GPU 1 |
| decompressed on | CPU 1 | GPU 1 | CPU 2 | GPU 2 | CPU 1 | GPU 1 | CPU 2 | GPU 2 |
| Tecnick dataset: 100 RGB images of $1200 \times 1200$ pixels | | | | | | | | |
| Ballé et al. (2018) | **0%** | 71% | 54% | 66% | 63% | 41% | 59% | 34% |
| ditto, integer prior | **0%** | **0%** | **0%** | **0%** | **0%** | **0%** | **0%** | **0%** |
| CLIC dataset: 2021 RGB images of various pixel sizes | | | | | | | | |
| Ballé et al. (2018) | **0%** | 78% | 68% | 78% | 77% | 52% | 78% | 54% |
| ditto, integer prior | **0%** | **0%** | **0%** | **0%** | **0%** | **0%** | **0%** | **0%** |

CPU 1: Intel Xeon E5-1650 | GPU 1: NVIDIA Titan X (Pascal)
CPU 2: Intel Xeon E5-2690 | GPU 2: NVIDIA Titan X (Maxwell)

Table 1: Decompression failure rates due to floating point round-off error on Tecnick and CLIC image datasets. When compressing and decompressing on the same CPU platform (first column), the Ballé et al. (2018) model decompresses all images correctly. However, when compressing on a GPU or decompressing on a different platform, a large percentage of the images fail to be decoded correctly. Implementing the prior of the same model using integer networks ensures correct decompression across all tested platforms.

during training, where $u$ is the input to the last layer and $/$ represents elementwise floating point division, and

$$y = Q\big((Hu + b)/c - o\big) \tag{19}$$

during evaluation. This can be rewritten strictly using integer arithmetic as:

$$y = \big(Hu + b - Q(o \odot c)\big) \oslash c, \tag{20}$$

where $\odot$ represents elementwise multiplication, and the rounded product can be folded into the bias $b$ as an optimization. This way, the representation is computed deterministically during evaluation, while during training, the marginal still resembles a smooth function, such that no regularization of the prior is necessary.

## 6 EXPERIMENTAL RESULTS

In order to assess the efficacy of integer networks to enable platform-independent compression and decompression, we re-implemented the image compression model described in Ballé et al. (2018),

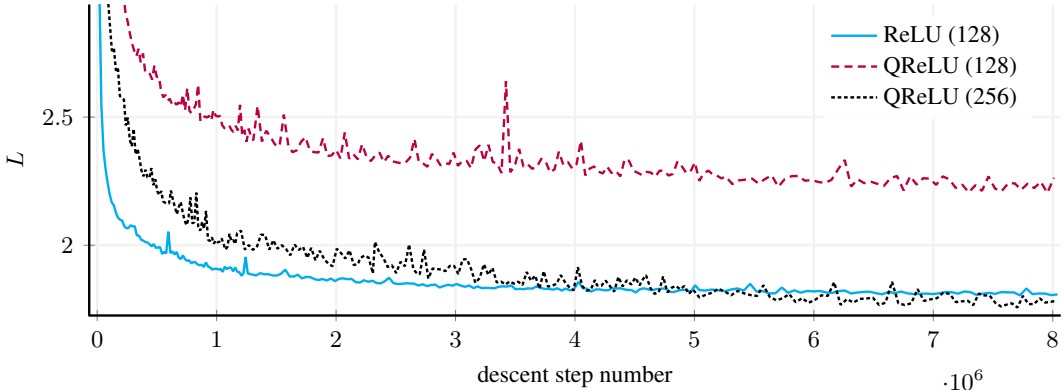

Figure 4: Loss function across training of Ballé (2018) model, evaluated on Kodak (1993), corresponding to the rate point at approximately 0.7 bits per pixel in figure 3, right panel. Generally, training of integer models takes somewhat longer and is somewhat noisier than training of floating point models. When matching floating point and integer networks for asymptotic performance (128 vs. 256 filters, respectively), integer networks take longer to converge (likely due to their larger number of filters). When matching by number of filters (128), it appears that the training time to convergence is about the same, but the performance ends up worse.

which is defined with a hyperprior. We compare the original model with a version in which the network $h_s$ computing the prior is replaced with an integer network. We used the same network architectures in terms of number of layers, filters, etc., and the same training parameters as in the original paper. The rate–distortion performance of the model was assessed on Kodak (1993) and is shown in figure 3 (left). The modified model performs identically to the original model, as it maps out the same rate–distortion frontier. However, it is much more robust to cross-platform compression and decompression (table 1). We tested compression and decompression on four different platforms (two CPU platforms and two GPU platforms) and two different datasets, Tecnick (Asuni and Giachetti, 2014) and CLIC (2018). The original model fails to correctly decompress more than half of the images on average when compression and decompression occurs on different platforms. The modified model brings the failure rate down to 0% in all cases.

It should be noted that the decreased accuracy of integer arithmetic generally leads to a lower approximation capacity than with floating point networks. We found that when implementing the models described in Ballé (2018) using integer networks throughout, the rate–distortion performance decreased (figure 3, right). The loss in approximation capacity can be compensated for by increasing the number of filters per layer. Note that this appears to increase the training time necessary for convergence (figure 4). However, note that increasing the number of parameters may not necessarily increase the size of the model parameters or the runtime, as the storage requirements for integer parameters (kernels, biases, etc.) are lower than for floating point parameters, and integer arithmetic is computationally less complex than floating point arithmetic in general.

## 7 DISCUSSION

There is a large body of recent research considering quantization of ANNs mostly targeted at image recognition applications. Courbariaux et al. (2015) train classification networks on lower precision multiplication. Hubara et al. (2016) and Rastegari et al. (2016) perform quantization down to bi-level (i.e., 1-bit integers) at inference time to reduce computation in classification networks. More recently, Wu et al. (2018) and others have used quantization during training as well as inference, to reduce computation on gradients as well as activations, and Baluja et al. (2018) use non-uniform quantization to remove floating point computation, replacing it completely with integer offsets into an integer lookup table.

While the quantization of neural networks is not a new topic, the results from the above techniques focus almost exclusively on classification networks. Denton et al. (2014), Han et al. (2016), and others have demonstrated that these types of networks are particularly robust to capacity reduction.

Models used for image compression, like many generative models, are much more sensitive to capacity constraints since they tend to underfit. As illustrated in Ballé (2018) and in figure 3 (right), this class of models is much more sensitive to reductions of capacity, both in terms of network size and the expressive power of the activation function. This may explain why our experiments with post-hoc quantization of network activations have never yielded competitive results for this class of model (not shown).

As illustrated in figure 1 and table 1, small floating point inconsistencies in variational latent-variable models can have disastrous effects when we use range coding to employ the models for data compression across different hardware or software platforms. The reader may wonder whether there exists other entropy coding algorithms that can convert discrete latent-variable representations into a binary representation, and which do not suffer from a sensitivity to perturbations in the probability model. Unfortunately, such an algorithm would always produce suboptimal results for the following reason. The source coding theorem (Shannon, 1948) establishes a lower bound on the average length of the resulting bit sequences, which range coding achieves asymptotically (i.e. for long bit sequences). The lower bound is given by the cross entropy between the marginal and the prior:

$$\mathbb{E}_{\boldsymbol{y} \sim m}[|b(\boldsymbol{y})|] \geq \mathbb{E}_{\boldsymbol{y} \sim m}[-\log_2 p(\boldsymbol{y} \mid \boldsymbol{\theta})], \tag{21}$$

where $|b(\boldsymbol{y})|$ is the length of the binary representation of $\boldsymbol{y}$. If an entropy coding algorithm tolerates error in the values of $p(\boldsymbol{y} \mid \boldsymbol{\theta})$, this means it must operate under the assumption of identical probability values for a range of values of $\boldsymbol{\theta}$ – in other words, discretize the probability values. Since the cross entropy is minimal only for $p(\boldsymbol{y} \mid \boldsymbol{\theta}) = m(\boldsymbol{y})$ (for all $\boldsymbol{y}$), this would impose a new lower bound on $|b(\boldsymbol{y})|$ given by the cross entropy with the discretized probabilities, which is greater or equal to the cross entropy given above. Thus, the more tolerant the entropy coding method is to errors in $p$, the further it deviates from optimal performance. Moreover, it is hard to establish tolerance intervals for probability values computed with floating point arithmetic, in particular when ANNs are used, due to error propagation. Hence, it is generally difficult to provide guarantees that a given tolerance will not be exceeded. For similar reasons, current commercial compression methods model probabilities exclusively in the discrete domain (e.g., using lookup tables; Marpe et al., 2003).

Our approach to neural network quantization is the first we are aware of which specifically addresses non-deterministic computation, as opposed to computational complexity. It enables a variety of possible variational model architectures and distributions to be effectively used for platform-independent data compression. While we aren't assessing its effects on computational complexity here, it is conceivable that complexity reductions can also be achieved with the same approach; this is a topic for future work.

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
