# OpenReview forum: "Integer Networks for Data Compression with Latent-Variable Models"
_ICLR.cc/2019/Conference_

### Official Review · AnonReviewer1 · 2018-11-01
**An interesting approach for a very important problem**

**Rating:** 8
**Confidence:** 3

**Review:**

The paper presents a very important problem of utilizing a model on different platforms with own numerical round-offs. As a result, a model run on a different hardware or software than the one on which it was trained could completely fail due to numerical rounding-off issues. This problem has been considered in various papers, however, the classification task was mainly discussed. In this paper, on the other hand, the authors present how the numerical rounding-off issue could be solved in Latent-Variable Models (LVM).

In order to cope with the numerical rouding-off issue, the authors propose to use integer networks. They consider either quantized ReLU (QReLU) or quantized Tanh (Qtanh). Further, in order to properly train the integer NN, they utilize a bunch of techniques proposed in the past, mainly (Balle, 2018) and (Balle et al., 2018). However, as pointed out in the paper, some methods prevent training instabilities (e.g., Eqs. 18 and 19). All together, the paper tackles very important problem and proposes very interesting solution by bringing different techniques proposed for quantized NNs together .

Pros:
+ The paper is well-written.
+ The considered problem is of great importance and it is rather neglected in the literature.
+ The experiments are properly carried out.
+ The obtained results are impressive.

Cons:
- A natural question is whether the problem could be prevented by post-factum quantization of a neural network. As pointed out in the Discussion section, such procedure failed. However, it would be beneficiary to see an empirical evidence for that.
- It would be also interesting to see how a training process of an integer NN looks like. Since the NN is quantized, instabilities during training might occur. Additionally, its training process may take longer (more epochs) than a training of a standard (float) NN. An exemplary plot presenting a comparison between an integer NN training process and a standard NN training process would be highly appreciated.
- (Minor remark). The paper is well-written, however, it would be helpful to set the final learning algorithm. This would drastically help in reproducibility of the paper.

--REVISION--
After reading the authors response and looking at the new version of the paper I decided to increase my score. The paper tackles very important problem and I strongly believe it should be presented during the conference.

---

> ### Author Response · Authors · 2018-11-19
> **Response to AnonReviewer1**
>
> We thank the reviewer for their notes and constructive comments.
>
> Regarding the remarks made:
>
> - We considered a post-hoc quantization approach, but did not end up experimenting with it much due to the complexity of its implementation. Simply quantizing the kernels and activations to an arbitrary choice of global quantization step size yielded terrible results (not even close to competitive – the model simply failed). We believe a solution which tries to maximize usage of the dynamic range of both the kernel coefficients and the activations, such as ours, can perform better, because it minimizes the quantization error throughout the network. In a post-training quantization approach, the actual range of activations would need to be measured empirically, and then a solution to a set of complex constraints would need to be found. We did not attempt this, because our goal was to come up with a method that requires only a minimum of post-training modifications to the model.
>
> - We agree that a training plot is a useful addition to the paper and have added one. We found that integer networks tend to train somewhat slower than floating point networks. When matching floating point and integer networks for asymptotic performance, integer networks take longer to converge (likely due to their larger number of filters). When matching by number of filters, it appears that the training time to convergence is roughly the same, but the performance ends up worse, of course. Indeed, the training loss also appears somewhat noisier.
>
> - We will try to add a summary of the method to the final paper.

---

> > ### Comment · AnonReviewer1 · 2018-11-21
> > **Response to authors**
> >
> > I would like to thank authors for their reply.
> >
> > 1) Post-hoc quantization
> > I understand your explanation and it sounds reasonably.
> >
> > 2) Training plot
> > Thank you very much for adding the plot. In my opinion the plot improves readability of the paper and confirms the intuition.
> >
> > 3) Summary of the method
> > It would be highly appreciated.
> >
> > Conclusion:
> > In my opinion the paper is very important and should be accepted. The problem of compression is extremely important from the practical point of view. I would even dare to claim that without a good set of tools for compression we will make very little progress in AI. As a result, I decided to improve my score.

---

### Official Review · AnonReviewer3 · 2018-11-06
**Application paper: how to modify latent variable models s.t. they survive range coding transmission.**

**Rating:** 7
**Confidence:** 3

**Review:**

This paper explains that range coding as a mechanism for transmitting latent-variable codes from source to target for decoding is severely sensitive to floating point errors.

The authors propose what amounts to an integer version of Balle 2018, and demonstrate that it allows for transmission between platforms without catastrophic errors due to numerical round-off differences.

The paper (and problem) is of low significance, but the authors present a neat solution.

Pros:
- Well defined problem and solution.
- Practical question relating to use of ANNs for data en/de-coding.

Cons:
- Presentation needs brushing up: e.g. why give two examples for H, b, v bit widths?
- Some approximations are not well motivated or justified.  E.g. why is it valid to replace gradient of a function that has 0 gradients with the identity?
- Paper needs some rewriting for clarity.  E.g. where is the kernel K defined?
- Lack of experimentation to justify the fact that the construction of (16) leads to instabilities, and is therefore less suitable than the method outlined here.

---

> ### Author Response · Authors · 2018-11-19
> **Response to AnonReviewer3**
>
> We thank the reviewer for their notes and constructive comments. However, we fundamentally do not understand the reasoning for the rejection. We think that all of the “cons” provided represent minor issues which can be fixed, and we address them further below.
>
> As such, only the claim that our paper is of “low significance” remains as a potential reason for the rejection. We disagree with that assessment, for the following reasons:
>
> - Application-orientedness: Indeed, our paper addresses a problem relevant to using variational models for compression, which is an “application” of neural networks. However, numerous “application” papers are submitted to ICLR every year. We do not think that compression as such is any less important than, say, natural language processing.
>
> - Generality: More importantly, we are not presenting “yet another compression method”. Our method for achieving machine determinism is relevant and applicable to all latent variable models. This includes some of the state of the art models for learned compression.
>
> - Optimality: We cannot guarantee that our method is optimal, or the only solution to this problem, for that matter. This is due to the nature of the problem (optimizing fully discrete models is hard). However, as far as we know, this problem has never been addressed before in a learning context or in this generality. It does not represent incremental work, but a qualitative contribution. The fact that there now exists a first documented solution to this problem, which can be used as a baseline for future work, is a strong point for qualifying our paper.
>
> We hope that the reviewer will respond to this and clarify what exactly is the reason for their rejection. We do not think we understand the reasoning at this point.
>
> Regarding the other points made:
>
> - We give two examples for the bit widths, because they show how the relevant parameters can be chosen in conjunction with different activation functions. We talk about this in the following paragraphs. We do not understand why the reviewer thinks that giving two examples for something constitutes bad quality.
>
> - Replacing the gradient of a quantization function (i.e. round()) with the identity is not a new idea (see, for example, Theis (2017), which we cite, among many others). If the reviewer would like a justification, how about the following:
> Taking the gradient of round() yields a sum of Dirac delta distributions, i.e.,
> d/dx round(x) = sum_i delta(x-i). Obviously, this gradient function is not going to be helpful for optimization, because it will produce too much variance in the gradients (+infinity for the half-integer positions, and 0 everywhere else). However, a smoothed version of it is helpful. If we convolve this gradient function with a triangle function (https://en.wikipedia.org/wiki/Triangular_function, which can be seen as the generator function for linear splines) to smooth it, the result is a constant (1), which corresponds to the identity function when used in backpropagation.
>
> - K is not a kernel, but the bit-width of the kernel (the kernel itself is defined as H). K is defined right after equation (7), where it is first mentioned. We modified the language to distinguish better between the kernel and its bit width. However, we believe that given sufficient time and care, any reader should have been able to distinguish this from the context.
>
> - We have to push back on the claim that we did not conduct enough experiments leading up to our conclusion that (16) causes instabilities. In fact, our initial goal was to simply reuse (16), since it had been already published and is simpler. However, the results were consistently worse, and on further inspection, we observed that the prior and the encoder would always end up in a kind of oscillatory behavior. We will try to find a visualization of this for the final paper. Theis (2017) used Gaussian scale mixtures centered on 0 as the prior, which is a form of regularization and might have helped reduce this, as it would prevent arbitrary shifts. However, we would like to be able to use arbitrarily powerful priors, as is generally the goal in variational approximation. The combination of (17) and (18) is unfortunately slightly less simple to implement, but it has the benefit that the prior distribution need not be subjected to regularization, and thus enables a more general solution. So, this choice was not only informed by empirical results, but also by the desire to be able to do without regularization.

---

> > ### Comment · AnonReviewer3 · 2018-11-27
> > **Response to author's rebuttal**
> >
> > Many thanks to the authors for their detailed response.
> >
> > I will summarize my reasoning in broad strokes, before delving in detail into the authors rebuttal.
> >
> > # Reasoning behind why (to me) the paper was not yet ready for publication:
> > 1. There are some well known good reasons for integer networks (typically based on speed/memory boosts), however I didn’t feel that the paper made it clear in the exposition why we should A: use range/arithmetic coding for VAE code transmission machine-machine (if we are sending elements from the latent space and not raw images, is this still more optimal than any alternative which would not suffer from such catastrophic FPE?).  B: why this shortcoming of range coding for floating point elements should be solved in the model, and not at a lower level—how are floating point numbers usually compressed and transmitted without loss, do none of these ad hoc fixes work?
> > 2. Some shortcomings in the exposition (again, to me).  I think it would be very useful for the reader if the authors provided some more background on what the transmission coding schemes are, and why we need to go this route (integer networks) to address their shortcomings.  This is what I was attempting to allude to when referring to the two example tables—I felt that some of this space could be used more effectively.  I am willing to accept that this (point 2) *could* be done in time for the camera ready.
> >
> > >- We give two examples for the bit widths, because they show how the relevant parameters can be chosen in conjunction with >different activation functions. We talk about this in the following paragraphs. We do not understand why the reviewer thinks >that giving two examples for something constitutes bad quality.
> > - As explained above, I do not think that this constitutes bad quality, I was merely suggesting that you could drop some redundancy and replace it with a more in depth analysis of alternatives to the channel coding problem.
> >
> > >- Replacing the gradient of a quantization function (i.e. round()) with the identity is not a new idea (see, for example, Theis >(2017), which we cite, among many others). If the reviewer would like a justification, how about the following:
> > >Taking the gradient of round() yields a sum of Dirac delta distributions, i.e.,
> > >d/dx round(x) = sum_i delta(x-i). Obviously, this gradient function is not going to be helpful for optimization, because it will >produce too much variance in the gradients (+infinity for the half-integer positions, and 0 everywhere else). However, a >smoothed version of it is helpful. If we convolve this gradient function with a triangle function (https://en.wikipedia.org/wiki/>Triangular_function, which can be seen as the generator function for linear splines) to smooth it, the result is a constant (1), >which corresponds to the identity function when used in backpropagation.
> > - OK, makes sense—but you do not cite this in the section where you say you ‘replace the derivative of Q with the identity function’, as an example.
> >
> > >- K is not a kernel, but the bit-width of the kernel (the kernel itself is defined as H). K is defined right after equation (7), where >it is first mentioned. We modified the language to distinguish better between the kernel and its bit width. However, we believe >that given sufficient time and care, any reader should have been able to distinguish this from the context.
> > - This is a little unfortunate, looking back in the revisions you can see that you previously state: “Here, we simply rescale each element of b′ using the bit-width of the kernel K, and round it to the nearest integer”.  Whilst I admit that you are correct that I could, given sufficient time, have determined that K was not some un-mentioned kernel, but instead the bit-width, I suspect I am not the only one to be thrown (at least temporarily) by that statement.
> >
> > Overall, based on the authors feedback and promises to address some of the issues, I would be willing to raise my current rating by 1.  To improve my score past the ‘accept’ barrier, however, I would want to be confident that points 1 + 2 (above) would be addressed in the camera ready.  Addressing 2. seems plausible, perhaps the authors can give some feedback on 1?  Alternatively, if there is no further feedback but the AC feels that point 1. does not matter, then the results (conditional on the importance of the problem-solution pair) are reasonably compelling.

---

> > > ### Author Response · Authors · 2018-11-28
> > > **Clarification**
> > >
> > > Thanks for making an effort to explain your concerns better. We believe that these are relatively easy to sort out, because it should be only a matter of presentation. We kept the introduction to the paper relatively terse, believing that it should provide enough motivation as is. We see now that we were assuming more familiarity with the compression literature than we maybe should. However, we can be more specific in the camera-ready version.
> > >
> > > There is currently no alternative to range coding or related methods. The reasoning is as follows:
> > >
> > > In order to losslessly transmit any data point y, we would ideally like to represent it as a sequence of bits whose length corresponds to its self-information under a model prior p shared by sender and receiver (-log_2 p(y)). This is compression in a nutshell: Likely data points under the prior will be represented with short sequences, unlikely ones with long sequences. Simply sending binary representations of floating point numbers not only fixes the lengths of the sequences to a constant, it also disregards the probability structure of the data points, which is suboptimal. The only known “entropy coding” algorithms which are applicable to arbitrary priors and achieve this mapping to bit sequences in an asymptotically optimal way are Huffman coding on one hand, and arithmetic/range coding (or ANS, all related) on the other. While Huffman coding is computationally less expensive in certain situations, it is generally not as practical with complex or conditional priors as the other class of algorithms, which is why we focus on range coding. Note that all entropy coding methods suffer from being sensitive to discrepancies in the prior between sender and receiver. This is a fundamental issue, because the optimal length of the sequence (-log_2 p(y)) and the likelihood of y under the prior (p(y)) are essentially the same quantity. We would also prefer to have a more “robust” range coder, such that small floating point discrepancies do not lead to catastrophic decoding errors, but such a method doesn’t exist. If we could invent one, it would invariably be suboptimal in terms of compression rate, because by design, it would need to allow for a certain error in p(y), which would have a direct effect on the sequence length. Furthermore, we would need to be able to control the error. Unfortunately, the magnitude of discrepancies in floating point computations can vary widely and is hard to predict. Therefore, all real-world image compression methods, including commercial ones, compute priors with discrete math (and all state-of-the-art methods use a form of range coding). There are no known short-cuts or ad-hoc fixes to this problem. It is well known in the compression community.
> > >
> > > Bringing this discussion into the latent-variable domain, we now sample from the encoder distribution e(y|x), where x is the data point, and then losslessly encode the latent representation y. The ideal length of the binary sequence is log_2 e(y|x) - log_2 p(y), with y sampled from e(y|x). This is again the self-information of y under a shared prior (-log_2 p(y)). It is possible, albeit practically difficult, to discount for the uncertainty in the encoder via bits-back coding (getting back -log_2 e(y|x) bits). However, methods to do that build on the previous class of entropy coding algorithms and suffer from the same problem. They also need the receiver to decode y, and then re-evaluate the encoder, and hence require not only the prior to be deterministic, but also the encoder and the decoder. Integer networks can also be applied here.
> > >
> > > We hope that this clarifies point 1 as well as point 2, and we’ll be happy to update the paper to include this reasoning (most likely as part of the introduction). If the reviewer thinks it is worthwhile, we can also include the motivation for the replacement gradient.

---

> > > > ### Comment · AnonReviewer3 · 2018-12-08
> > > > **Response to clarification**
> > > >
> > > > Ok, thank you for the clarification; I am not familiar enough with the coding literature to be confident that Huffman would be so much worse for this particular task (I follow the reasoning but not sure how complex the latent structure would have to be), but I’ll take your word for it.
> > > >
> > > > As you will/have included the explanation in the paper, I am happy to improve my score to above the accept threshold.

---

> > > > > ### Author Response · Authors · 2018-12-10
> > > > > **Thank you**
> > > > >
> > > > > Thank you for updating your score.
> > > > >
> > > > > Note that Huffman coding with a conditional probability model would have equivalent issues, because the design of the Huffman code would again be very sensitive to fluctuations in the probabilities. It's a fundamental issue with entropy coding methods in general.
> > > > >
> > > > > We'll think about how we can improve this explanation further, to try to make it as clear as possible for the final paper. We believe it will benefit the community to have everyone on the same page.

---

> > > ### Author Response · Authors · 2018-12-07
> > > **Feedback**
> > >
> > > Dear AnonReviewer3,
> > >
> > > we hope that the feedback we provided clarifies the issue. We'd be happy to go into more detail if necessary. Please let us know if you have any further questions or concerns.
> > >
> > > Thank you.

---

### Official Review · AnonReviewer4 · 2018-11-11
**Interesting read; would be helpful to better explain difficulties in training**

**Rating:** 6
**Confidence:** 3

**Review:**

This well-written paper addresses the restrictions imposed by binary communication channels on the deployment of latent variable models in practice. In order to range code the (floating point) latent representations into bit-strings for practical data compression, both the sender and receiver of the binary channel must have identical instances of the prior despite non-deterministic floating point arithmetic across different platforms. The authors propose using neural networks that perform integer arithmetic (integer networks) to mitigate this issue.

Pros:
- The problem statement is clear, as well as the approach taken to addressing the issue.
- Section 5 did a nice job tying together the relevant literature on using latent variable models for compression with the proposed integer network framework.
- The experimental results are good; particularly, Table 1 provides a convincing case for how using integer networks remedies the issue of decompression failure across heterogeneous platforms.

Cons:
- In Section 3, it wasn’t clear to me as to why the authors were using their chosen gradient approximations with respect to H’, b’ and c’. Did they try other approximations but empirically find that these worked best? Where did the special rescaling function s come from? Some justifications for their design choices would be appreciated.
- The authors state in Section 2 that the input scaling is best determined empirically -- is this just a scan over possible values during training? This feels like an added layer of complexity when trying to train these networks. It would be nice if the authors could provide some insight into exactly how much easier/difficult it is to train integer networks as opposed to the standard floating point architectures.
- In Section 6, the authors state that the compromised representational capacity of integer networks can be remedied by increasing the number of filters. This goes back to my previous point, but how does this “larger” integer network compare to standard floating point networks in terms of training time?

---

> ### Author Response · Authors · 2018-11-19
> **Response to AnonReviewer4**
>
> We thank the reviewer for their notes and constructive comments.
>
> Regarding the questions raised:
> - Indeed, the approximations we present here are the end result of a long list of experiments. Our experiments included:
>   - Several different gradient substitutes such as straight-through, tanh, etc.
>   - Replacing the gradient of the activation functions vs. adding uniform noise as a substitute of the quantization during training.
>   - A distillation approach (i.e. first training a floating point compression model, and then attempting to match an integer model to it in terms of a simpler loss function).
>   - We considered a post-hoc quantization approach, but did not end up experimenting with it much due to the complexity of its implementation.
>
> - We found that the presented solution is the best performing, while being conceptually simple and relatively easy to implement. Of course, we cannot claim that this solution is optimal in any sense. Optimizing fully discrete models is a hard problem. To our knowledge, our method is the first attempt to solve the non-determinism problem of neural networks in this context, and certain choices were made in an ad-hoc way. However, we hope that it can be used as a baseline for improved methods in the future.
>
> - We agree we should have been more elaborate explaining how some of the choices were made. We improved the paper in this regard.
>   - The rescaling function s is designed to maximize usage of the dynamic range of kernel coefficients. It simply rescales the kernels such that the minimum or maximum coefficient of each filter is identical to one of the dynamic range bounds, but keeps zero at zero.
>   - The function r is a reparameterization of the divisor c, such that for small values of it, the effective descent step size on it is reduced. When c is small, large perturbations in c can lead to excessively large fluctuations of the quotient (i.e., the input to the nonlinearity). This leads to instabilities in training. The reparameterization ameliorates this. The effective step size on c ends up being multiplied with a factor that is approximately linear in c (this is previous work; see Ballé, 2018).
>   - We did not conduct a lot of experiments regarding the choice of input scaling. We believe it doesn’t matter too much; in our experiments, we simply chose a value “by eye”, so that the tanh nonlinearity is reasonably well represented with the lookup table. (I.e., in figure 2, right panel, we made sure that at least two or three input values are mapped to each output value, in order to preserve the approximate shape of the nonlinearity.)
>
> - We found in all of our experiments that integer networks tend to train somewhat slower than floating point networks. When matching floating point and integer networks for asymptotic performance, integer networks take longer to converge (likely due to their larger number of filters). When matching by number of filters, it appears that the training time is about the same, but the performance ends up worse, of course. We have added a figure to the paper to show this.

---

### Author Response · Authors · 2019-02-22
**Paper revision uploaded**

We uploaded the finalized revision of the paper, with an added paragraph in the discussion justifying our approach, and addressing the issues raised by AnonReviewer3.

We hope that the explanation given is easy to follow, and improves the presentation of our paper.

Thank you, and looking forward to meeting in New Orleans!

---

### Meta-Review · Area_Chair1 · 2018-12-13
**Interesting solution to a practical problem**

**Confidence:** 4
**Recommendation:** Accept (Poster)

**Metareview:**

This paper addresses the issue of numerical rounding-off errors that can arise when using latent variable models for data compression,  e.g., because of differences in floating point arithmetic across different platforms (sender and receiver). The authors propose using neural networks that perform integer arithmetic (integer networks) to mitigate this issue. The problem statement is well described, and the presentation is generally OK, although it could be improved in certain aspects as pointed out by the reviewers. The experiments are properly carried out, and the experimental results are good.
Thank you for addressing the questions raised by the reviewers. After taking into account the author's responds, there is consensus that the paper is worthy of publication. I therefore recommend acceptance.